# Multi-pathology MRI lesion segmentation in a multi-centre cohort of patients with focal epilepsy: a MELD study

**Mathilde Ripart**[1]                                    M.RIPART@UCL.AC.UK

[1] *UCL Great Ormond Street Institute of Child Health, London, UK*

**MELD-consortium**                                    SEE APPENDIX

**Sophie Adler**[*1]                          SOPHIE.ADLER.13@UCL.AC.UK

**Konrad Wagstyl**[*2,1]                          KONRAD.WAGSTYL@GMAIL.COM

[2] *School of Biomedical Engineering & Imaging Sciences, KCL, London, UK*

**Editors:** Accepted for publication at MIDL 2024

## Abstract

Drug-resistant focal epilepsy can be caused by structural lesions for which surgery can be curative if the lesion is found and fully resected. We built a large multi-centre MRI dataset of 1181 patients with focal epilepsy and 1009 healthy controls to train a state-of-the-art nnU-Net model to segment a range of aetiologies on T1w MRI scans. The model was able to detect different aetiological causes of focal epilepsy with a sensitivity of 73% in patients and a specificity of 90% in controls.

**Keywords:** lesion detection, nnU-Net, MRI, epilepsy, multi-centre

## 1. Introduction

Drug-resistant focal epilepsy (DRFE) can be caused by a range of structural brain abnormalities, from large tumours to small, difficult-to-detect cortical malformations. Resective brain surgery is a potentially curative treatment. The likelihood of being offered surgery and the rate of post-surgical seizure freedom is higher if the abnormality is detected on MRI. Diagnosing these lesions is a widely-recognised challenge in epilepsy, with 10-50% of lesions described as "MRI negative", i.e. not found on MRI review (Eriksson et al., 2023; Téllez-Zenteno et al., 2010). AI models have been developed to aid in the detection of specific epilepsy pathologies, such as focal cortical dysplasia (FCD) or hippocampal sclerosis (HS) (Spitzer and Ripart et al., 2022; Ripart et al., 2023). However, at the point of MRI review, the underlying pathology is not known and there is an urgent clinical need to develop models capable of detecting a broad range of those pathologies. In this study, we leverage the MELD Focal Epilepsy dataset, the largest collection of 3D MRI scans in patients with drug-resistant focal epilepsy, to investigate whether a single classifier can segment a variety of structural causes of focal epilepsy.

## 2. Material and Methods

### 2.1. Cohort and dataset preparation

Following national and local Institutional Review Board approval, data were retrospectively collected and anonymised from 18 epilepsy centres worldwide. The study included 1181

---

* Contributed equally

patients with DRFE and a radiological or histopathological lesion. Pathologies included focal cortical dysplasia (FCD), hippocampal sclerosis (HS), low-grade epilepsy-associated tumours (LEAT) (including dysembryoplastic neuroepithelial tumours (DNET) and ganglogliomas), hypothalamic hamartoma (HH), cavernoma (CAV), polymicrogyria (PMG), periventricular nodular heterotopia (PNH) and other pathologies. A cohort of 1009 controls was included. All participants had a 3D T1w scan acquired on a 1.5T or 3T MRI scanner. The cohort was split into 80% train-val and 20% test datasets using stratified random sampling by pathology, site, and histological status to ensure a good representation of each of the groups in the training and test datasets (Figure 1). T1w scans underwent neuroanatomical brain segmentation using Synthseg (Billot et al., 2023). For patients with hippocampal lesions, the SynthSeg hippocampal segmentation was used as the ground truth lesion mask. For all other lesions, experts manually drew the ground truth lesion mask on the preoperative T1w scan. Lesion mask preprocessing included a dilation-erosion algorithm to fill in small defects/holes derived from masking in 2D and removal of any CSF voxels, using the Synthseg CSF segmentation. For patients with dual pathologies (e.g. HS & FCD in FCD Type 3a), hippocampal labels were merged with the manual cortical lesion mask.

## 2.2. Training and evaluating the model

T1w MRI scans were used to train a nnU-Net model (Isensee et al., 2021) to segment focal lesions using the ground truth lesion masks. The nnU-Net was trained for 1000 epochs with the '3d_fullres' configuration. The model's predictions were connected to regroup any fragmented clusters, discarding those containing fewer than 100 voxels. The model was evaluated on the test dataset for its sensitivity in detecting lesions (i.e. minimum 1 voxel overlap between the prediction and the mask) and specificity in controls (i.e. no prediction). We report the performance of the model on the test dataset and test dataset subgroups: pathology, age group, histology confirmation and MRI status.

## 3. Results

Model performance is detailed in Figure 1. The model detected 170 out of the 232 focal epilepsy abnormalities in patients (73% sensitivity). It accurately detected 63% of FCD, 91% of HS, 80% of LEAT, 60% of CAV, 67% of PMG and 50% of HH, PNH and other pathologies. Notably, the model accurately detected 13 out of the 29 abnormalities (45%) previously reported MRI-negative. Patients had a median of 0 false-positive clusters predicted, and the model accurately predicted no putative lesions in 182 out of 202 healthy controls (90% specificity). Figure 2 depicts examples of seven accurate predictions in patients with different underlying pathologies. The radiological characteristics visibly differ between the pathologies. Nonetheless, the model was able to segment these pathologies with a good overlap with the manual lesion masks.

## 4. Discussion

We demonstrate that a single deep-learning model can segment a variety of pathologies associated with focal epilepsy on T1w MRI, on an heterogenous cohort, representing a range

of ages, countries and MRI scanners. The algorithm successfully detected 45% of MRI-negative lesions, evidence of its potential utility as a radiological adjunct. Furthermore, radiological delineation of subtle lesions in epilepsy typically relies on inspection of a 3D FLAIR, and we anticipate future work incorporating additional sequences will further boost model performance. This work represents a significant step forward in the development of a robust automated lesion segmentation tool that could help in the presurgical planning for patients with focal epilepsy, potentially aiding early identification and referral of candidates for epilepsy surgery, and reducing the need for invasive investigations.

| | | Demographic and clinical information | | Model performances on the test dataset | |
|---|---|---|---|---|---|
| | | Training dataset | Test dataset | Detection rate % (n) | Number of FP cluster IQR [0.25, 0.5, 0.75] |
| Healthy controls (n) | | 807 | 202 | 90.1% (182/202) | [0, 0, 0] |
| Sex (m:f) | | 319:488 | 70:132 | - | - |
| Age at scan, years (median [min-max]) | | 34.0 [0.4-78.0] | 36.5 [1.9-68.0] | - | - |
| Patients (n) | | 949 | 232 | 73.3% (170/232) | [0, 0, 1] |
| Sex (m:f) | | 486:463 | 118:114 | - | - |
| Age of epilepsy onset, years (median [IQR]) | | 7.5 [2.5-15.0] | 7.5 [3.0-15.0] | - | - |
| Age at scan | years (median [min-max]) | 22.5 [0.1-69.0] | 22.0 [1.0-60.4] | - | - |
| | Children (< 18 years old) | 402 | 96 | 64.6% (62/96) | [0, 0.5, 2 ] |
| | Adults (≥ 18 years old) | 547 | 136 | 79.4% (108/136) | [0, 0, 1] |
| Histopathology (n) | FCD | 389 | 101 | 63.4% (64/101) | [0, 1, 2] |
| | HS | 298 | 74 | 90.5% (67/74) | [0, 0, 0] |
| | LEAT | 124 | 30 | 80.0% (24/30) | [0, 0, 1] |
| | CAV | 45 | 10 | 60.0% (6/10) | [0, 0, 1] |
| | PMG | 16 | 3 | 66.7% (2/3) | [0, 0, 0] |
| | HH | 16 | 2 | 50.0% (1/2) | [2, 2, 2] |
| | PNH | 16 | 4 | 50.0% (2/4) | [0.75, 1, 1 ] |
| | Other | 45 | 8 | 50.0% (4/8) | [0, 0, 1] |
| Confirmed on histopathology | yes | 765 (81%) | 189 (81%) | 74.1% (140/189) | [0, 0, 1] |
| | no | 184 (19%) | 43 (19%) | 69.8% (30/43) | [0, 1, 1] |
| Ever reported MRI-negative (n) | yes | 93/722 (13%) | 29/176 (16%) | 44.8% (13/29) | [0, 0, 1] |
| | no | 629/722 (87%) | 147/176 (83%) | 72.1% (106/147) | [0, 0, 1] |

Figure 1: Demographic and clinical information of the training and test datasets and model performances on the test dataset. (FP: False Positives, IQR: interquartile range)

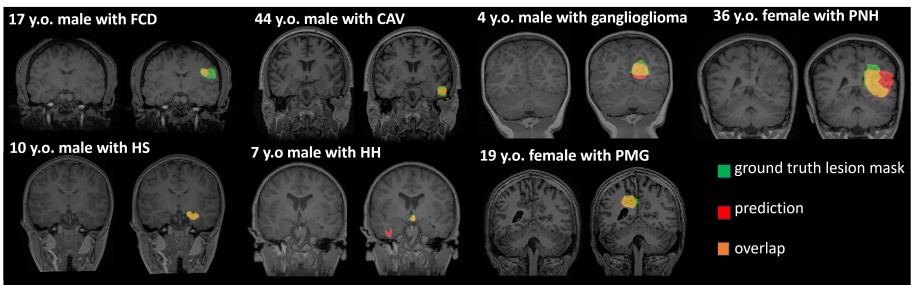

Figure 2: Examples of segmentation of multiple focal epilepsy pathologies. Coronal views of the T1w scan with and without the ground truth lesion mask, model prediction, and their overlap overlayed.

## Acknowledgments

MR and SA are supported by the Rosetrees Trust (A2665) and Epilepsy Research Institute (P2208). SG and HP are supported by Seed grant from Amrita Institute of Medical Science. PS and KH are supported by Health and Care Research Wales. NTC is supported by AAN Career Development Award. IW is supported by NIH R01 NS109439. GMR and AI are supported by FONDECYT (1210195, FONDECYT 1210176, 1220995). AR is supported by Supported by Research Program "MNESYS (PNRR-MUR-M4C2 PE0000006). MHE is supported by The Sigrid Jusélius Foundation. JD is supported by BRC. GPW is supported by MRC (G0802012, MR/M00841X/1). JSD is supported by NIHR. JEI is supported by NIH (1RF1MH123195, 1R01AG070988, 1R01EB031114, 1UM1MH130981, 1RF1AG080371). KW is supported by the Wellcome Trust.

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

## Appendix A. MELD consortium

**Siby Gopinath**[3]                                                                 sibygopi@gmail.com
[3] Amrita Advanced Centre for Epilepsy (AACE), Amrita Institute of Medical Sciences, Amrita Vishwa Vidyapeetham, Kochi, Kerala, India
**Harilal Parasuram**[3]                                                       harilal.navami@gmail.com
**Jiajie Mo**[4]                                                                       jiajiemo@foxmail.com
[4] Beijing Tiantan Hospital, Beijing, China
**Kai Zhang**[4]                                                                   zhangkai62035@sina.com
**Marcus Likeman**[5]                                                     marcus.likeman@uhbw.nhs.uk
[5] Bristol Royal Hospital for Children, Bristol, UK
**Georgian Ciobotaru**[6]                                        neurochirurgie@doctorciobotaru.ro

[6] Central Emergency Military Hospital, Bucarest, Romania

**Philip Sequeiros-Peggs**[7]      P.SEQUEIROSPEGGS@BTINTERNET.COM

[7] University Hospital of Wales, Cardiff, UK

**James Galea**[7]      GALEAJ2@CARDIFF.AC.UK

**Khalid Hamandi**[7]      HAMANDIK@CARDIFF.AC.UK

**Venkata Sita Priyanka Illapani**[8]      VILLAPANI@CHILDRENSNATIONAL.ORG

[8] Center for Neuroscience, Children's National Hospital, US

**Hua Xie**[8]      HXIE@CHILDRENSNATIONAL.ORG

**Nathan T. Cohen**[8]      NCOHEN@CHILDRENSNATIONAL.ORG

**William D. Gaillard**[8,9]      WGAILLAR@CHILDRENSNATIONAL.ORG

[9] Geo Washington University, US

**Ting-Yu Su**[10,11]      SUT@CCF.ORG

[10] Epilepsy Center, Neurological Institute, Cleveland Clinic, Cleveland, OH, US

[11] Biomedical Engineering, Case Western Reserve University, Cleveland, OH, US

**Ryuzaburo Kochi**[10]      RYUZABURO0618@HOTMAIL.CO.JP

**Irene Wang**[10]      WANGI2@CCF.ORG

**Gonzalo M. Rojas-Costa**[12,13]      GONZALO.ROJAS.COSTA@GMAIL.COM

[12] Advanced Epilepsy Center, Clínica las Condes, Santiago, Chile

[13] School of Medicine, Finis Terrae University, Santiago, Chile

**Agustín Ibáñez**[14,15]      AGUSTIN.IBANEZ@GBHI.ORG

[14] Latin America Brain Health Institute (BrainLat), Universidad Adolfo Ibanez, Chile

[15] Global Brain Health Institute (GBHI), Trinity College Dublin, Ireland

**Costanza Parodi**[16]      COSTANZA.BIANCA.PARODI@GMAIL.COM

[16] Department of Neuroradiology, IRCCS Istituto Giannina Gaslini, Genova, Italy

**Mariasavina Severino**[16]      MARIASAVINASEVERINO@GASLINI.ORG

**Domenico Tortora**[16]      DOMENICOTORTORA@GASLINI.ORG

**Giulia Nobile**[17]      GIULIA.NOBILE@GASLINI.ORG

[17] Unit of Child Neuropsychiatry, Department of Medical and Surgical Neuroscience and Rehabilitation, IRCCS Istituto Giannina Gaslini, Genova, Italy

**Alessandro Consales** [18]      ALESSANDROCONSALES@GASLINI.ORG

[18] Division of Neurosurgery, IRCCS Istituto Giannina Gaslini, Genova, Italy.

**Antonella Riva**[19,20]      RIVA.ANTO94@GMAIL.COM

[19] IRCCS Istituto Giannina Gaslini, Genova, Italy

[20] Department of Neurosciences, Rehabilitation, Ophthalmology, Genetics, Maternal and Child Health, University of Genova, Genova, Italy

**Felice D'Arco**[21]      FELICE.D'ARCO@GOSH.NHS.UK

[21] Radiology department, Great Ormond Street Hospital for Children, London, UK

**Kshitij Mankad**[21]      KSHITIJ.MANKAD@GOSH.NHS.UK

**Aswin Chari**[22]      ASWIN.CHARI.18@UCL.AC.UK

[22] Great Ormond Street Hospital for Children, London, UK

**Martin Tisdall**[22]      MARTIN.TISDALL@GOSH.NHS.UK

**Maria H. Eriksson**[1,23]      M.ERIKSSON.16@UCL.AC.UK

[23] The Hospital for Sick Children (SickKids), Toronto, Canada

**Rory J. Piper**[1,24]      RORY.PIPER.20@UCL.AC.UK

[24] Department of Neurosurgery, Great Ormond Street Hospital, London, UK

**Chris A. Clark**[1]                                    CHRISTOPHER.CLARK@UCL.AC.UK
**J. Helen Cross**[1]                                    H.CROSS@UCL.AC.UK
**Torsten Baldeweg**[1]                                  T.BALDEWEG@UCL.AC.UK
**Adrián Valls Carbó**[25]                               ADRIANVALLSC@GMAIL.COM
[25] Department of Neurology, Epilepsy Program, Ruber International Hospital, Madrid, Spain
**Rafael Toledano**[25]                                  RTOLEDANO@RUBERINTERNACIONAL.ES
**Callum M. Simpson**[26]                                C.SIMPSON5@NEWCASTLE.AC.UK
[26] School of Computing, Newcastle University, Newcastle upon Tyne, UK
**Yujiang Wang**[26]                                     YUJIANG.WANG@NEWCASTLE.AC.UK
**Peter Taylor**[26]                                     PETER.TAYLOR@NEWCASTLE.AC.UK
**Antonio Napolitano**[27]                               ANTONIO.NAPOLITANO@OPBG.NET
[27] Medical Physics Unit, Bambino Gesù children's hospital, IRCCS, Rome, Italy
**Luca De Palma**[28]                                    LUCA.DEPALMA@OPBG.NET
[28] Neurology, Epilepsy and Movement Disorders, Bambino Gesù Children's Hospital, IR-CCS, Rome, Italy
**Alessandro De Benedictis**[29]                         ALESSANDRO.DEBENEDICTIS@OPBG.NET
[29] Neurosurgery Unit, Bambino Gesù Children's Hospital, IRCCS, Rome, Italy
**Maria Camilla Rossi-Espagnet**[30]                     MCAMILLA.ROSSI@OPBG.NET
[30] Functional and Interventional Neuroradiology Unit, Bambino Gesù children's hospital, IRCCS, Rome,Italy
**Anna Willard**[31]                                     ANNA.WILLARD@MONASH.EDU
[31] Department of Neuroscience, Monash University, Melbourne, Australia
**Ben Sinclair**[31,32]                                  BEN.SINCLAIR@MONASH.EDU
[32] Department of Neurology, Alfred Health, Melbourne, Australia
**Lucy Vivash**[31,32]                                   LUCY.VIVASH@MONASH.EDU
**Joshua Pepper**[33]                                    JOSHUA.PEPPER1@NHS.NET
[33] Birmingham Women's and Children's NHS Foundation Trust, Birmingham, UK
**Stefano Seri**[33]                                     S.SERI@NHS.NET
**David N. Vaughan**[34,35]                              DAVID.VAUGHAN@FLOREY.EDU.AU
[34] Florey Institute of Neuroscience and Mental Health, Melbourne, Australia
[35] Department of Neurology, Austin Health, Melbourne, Australia
**Donna Parker**[34]                                     DONNA.PARKER@FLOREY.EDU.AU
**Graeme Jackson**[34]                                   GJACKSON@BRAIN.ORG.AU
**Heath R. Pardoe**[34]                                  HEATH.PARDOE@FLOREY.EDU.AU
**Jane de Tisi**[36,37]                                  J.DETISI@UCL.AC.UK
[36] UCL Queen Square Institute of Neurology, London, UK
[37] National Hospital for Neurology and Neurosurgery, London, UK
**Gavin P. Winston**[36,38]                              GAVIN.WINSTON@QUEENSU.CA
[38] Department of Medicine, Queen's University, Kingston, Canada
**John S. Duncan**[36,37]                                J.DUNCAN@UCL.AC.UK
**Clarissa L. Yasuda**[39,40]                            CYASUDA@UNICAMP.BR
[39] UNICAMP University of Campinas, Campinas, Brasil
[40] Brazilian Institute of Neuroscience and Neurotechnology, Brazil
**Lucas Scárdua-Silva**[39]                              SCARDUA@UNICAMP.BR

**Marina K. M. Alvim**[39]                                   marinakm@unicamp.br

**Fernando Cendes**[39]                                     fcendes@gmail.com

**Lennart Walger**[41,42]                              lennart.walger@ukbonn.de

[41] Department of Neuroradiology, University Hospital Bonn, Bonn, Germany

[42] Department of Epileptology, University Hospital Bonn, Bonn, Germany

**Theodor Rüber**[41,42]                              theodor.Rueber@ukbonn.de

**Juan Eugenio Iglesias**[43,44]                          e.iglesias@ucl.ac.uk

[43] Massachusetts General Hospital & Harvard Medical School, USA

[44] Centre for Medical Image Computing, UCL, UK

