# OpenReview forum: "Multi-pathology MRI lesion segmentation in a multi-centre cohort of patients with focal epilepsy: a MELD study"
_MIDL.io/2024/Short_Papers — MIDL 2024 Short Papers_

### Official Review · Reviewer_Hs2p · 2024-04-18

**Confidence:** 4
**Final Rating:** 3.5

**Review:**

The study develops and evaluates a unet for segmentation of focal epilepsy lesions. Using a database of epilepsy patients and controls, it shows 73% of the lesions can be found by at least 1 pixel (sensitivity), and that the same model detects nothing in 90% of controls (precision).

Strenghts:
- The research hypothesis is clear (can we develop a model segmenting different types of focal epilepsy lesions?)
- The paper is well written

Weaknesses:
- It is difficult to judge the quality of the results, as only few metrics have been reported (sensitivity/ precision) and there is no indication if these results are meaningful / practically useful in the specific context of the application.

More detailed comments about the weakness:
- It is difficult to judge the quality of the results, as only few metrics have been reported (sensitivity/ precision) and there is no indication if these results are meaningful / practically useful in the specific context of the application. With respect to *detection accuracy* (whether the method can detect a lesion), is 73% sensitiivty anywhere close to what humans/clinicians have, therefore results anywhere close to useful? This is unclear (the authors did not that the method found some of the lesions missed by radiologists, but this is accompanied by missing ~30% of those found). In future extensions, I'd advise the authors to try improve on this, as it's the original motivation.

- Similar to the above, there is no metric that quantifies the quality of segmentation, e.g. DSC. The way sensitivity is defined is that a lesion is considered found (true positive) if even a single 1 pixel is segmented. This does not allow us to understand whether the tool could be useful as a segmentation tool for quantitative analysis of lesions (eg their volume).

Justification for rating:
The paper is well written and there is not too much work on this type of lesions, so eventhough I cannot extract too confident results about the quality of the model/results, the community may find it an interesting project and a step forward to discuss.

---

### Decision · Program_Chairs · 2024-04-26

Accept